# Vancomycin-Resistant *E. faecium*: Addressing Global and Clinical Challenges

**DOI:** 10.3390/antibiotics14050522

**Published:** 2025-05-19

**Authors:** Daniel E. Radford-Smith, Daniel C. Anthony

**Affiliations:** Department of Pharmacology, University of Oxford, Oxford OX1 3QT, UK

**Keywords:** vancomycin-resistant *Enterococcus faecium*, antimicrobial resistance, antimicrobial stewardship

## Abstract

Antimicrobial resistance (AMR) poses a profound threat to modern healthcare, with vancomycin-resistant *Enterococcus faecium* (VREfm) emerging as a particularly resilient and clinically significant pathogen. This mini-review examines the biological mechanisms underpinning VREfm resistance, including biofilm formation, stress tolerance, and the acquisition of resistance genes such as *vanA* and *vanB*. It also explores the behavioural, social, and healthcare system factors that facilitate VREfm transmission, highlighting disparities in burden across vulnerable populations and low-resource settings. Prevention strategies are mapped across the disease pathway, spanning primary, secondary, and tertiary levels, with a particular focus on the role and evolving challenges of antimicrobial stewardship programmes (ASP). We highlight emerging threats, such as rifaximin-induced cross-resistance to daptomycin, which challenge conventional stewardship paradigms. Finally, we propose future directions to enhance global surveillance, promote equitable stewardship interventions, and accelerate the development of innovative therapies. Addressing VREfm requires a coordinated, multidisciplinary effort to safeguard the efficacy of existing antimicrobials and protect at-risk patient populations.

## 1. Introduction

Antimicrobial agents are a cornerstone of modern medicine and society, enabling the treatment of infectious diseases, providing perioperative protection, and supporting patients with compromised immunity, such as those undergoing chemotherapy. However, antimicrobial resistance (AMR) is a natural and inevitable phenomenon, first observed shortly after the mass production of penicillin [1]. Although rapid antibiotic development during the 1950s–60s initially mitigated the threat, the widespread clinical use of antimicrobials, combined with a sharp decline in novel antibiotic discovery, has culminated in the current AMR crisis [2,3].

Among the pathogens of increasing concern are enterococci, a genus of Gram-positive bacteria implicated in a rising number of bloodstream infections (BSI). Particularly troubling is the emergence of *Enterococcus faecium* as the predominant species associated with enterococcal bacteraemia in England [4] and the United States [5]. *E. faecium* infections are more commonly associated with multidrug-resistant (MDR) phenotypes, including resistance to vancomycin and gentamicin [6]. Vancomycin-resistant *Enterococcus faecium* (VREfm) is now a significant pathogen in healthcare-associated infections, particularly in critical care settings, where it is associated with high mortality rates [7,8].

The persistence and spread of VREfm reflect not only biological resilience but also human behaviours and vulnerabilities within healthcare systems. Vulnerable populations, particularly the immunocompromised and critically ill, face disproportionate risks, exacerbating health inequalities. Here, we review the biological, behavioural, and systemic determinants of VREfm emergence and spread. By mapping prevention strategies across the disease pathway, with particular emphasis on the role of antimicrobial stewardship programmes (ASP), we provide an up-to-date synthesis of recent evidence. We also clarify emerging challenges, such as the discovery of novel cross-resistance mechanisms, shifting patterns of resistance genes, and disparities in global surveillance capacity. By integrating biological, clinical, and policy-level insights, this mini-review offers a multidisciplinary framework to inform future research, guide stewardship interventions, and strengthen the global response to VREfm.

## 2. Determinants of VREfm Infections and Resistance

### 2.1. Biological Factors

A key biological determinant of VREfm is its remarkable resilience and adaptability in hostile environments. *E. faecium* can survive for extended periods in nutrient-poor aqueous environments and on dry hospital surfaces, making it highly persistent in healthcare settings [9,10]. Although it does not form spores, it can enter dormant states under stress, evading antibiotics that target actively growing cells [11].

*E. faecium* is generally considered less proficient at biofilm formation than *E. faecalis*, but multiple studies have shown that clinical isolates can form biofilms under specific conditions [12]. Biofilm development in *E. faecium* involves a range of surface adhesins and regulatory proteins, including the ebpABC pilus operon [13,14], AtlA autolysin [15], and the Esp and Acm surface proteins [16,17]. Deletion of ebpABC or esp has been shown to reduce biofilm formation and virulence in models of urinary tract infection and endocarditis [13,16]. AtlA contributes to biofilm maturation by mediating the release of extracellular DNA (eDNA), a key structural component of the biofilm matrix [15]. The enterococcal biofilm regulator B (EbrB) modulates expression of esp and other surface proteins, and its deletion impairs biofilm formation and intestinal colonization [18]. Additional regulators such as AsrR, involved in antibiotic and stress responses, may repress biofilm formation in vivo, and deletion of asrR increases biofilm persistence in animal models [19]. Although less well characterised than in *E. faecalis*, biofilm formation in *E. faecium* appears to be strain-specific and influenced by environmental stress and host interactions. Notably, *E. faecium* isolates that harbour biofilm-associated genes—such as *esp*, *ebpABC*, and *atlA*—are more likely to exhibit resistance to multiple antibiotics, including aminoglycosides, suggesting that biofilm-related phenotypes may contribute to both persistence and antimicrobial resistance in clinical settings [20].

Resistance to β-lactam antibiotics in *E. faecium* is largely attributed to mutations in the *pbp5* gene, often found within a mobile chromosomal region, and strongly associated with the evolution of multidrug-resistant clade A1 lineages [21]. This clade harbours numerous resistance determinants, including genes for vancomycin resistance. Alarmingly, tolerance to last-resort antibiotics like linezolid and daptomycin is also increasing, often emerging rapidly during treatment [22,23,24].

Vancomycin remains a key therapeutic agent for managing serious infections caused by Gram-positive organisms, including *Enterococcus faecium* and methicillin-resistant *Staphylococcus aureus* (MRSA) [25]. Its antimicrobial effect depends on binding to the D-Ala-D-Ala termini of peptidoglycan precursors, thereby disrupting bacterial cell wall synthesis. Resistance in *E. faecium* typically arises through acquisition of the *vanA* or *vanB* gene clusters, which alter the target of vancomycin. The *vanA* cluster leads to high-level resistance to both vancomycin and teicoplanin, whereas *vanB* confers resistance to vancomycin alone and with more variable expression levels [26]. These resistance genes are often encoded on the transposable element *Tn1546*, which includes regulatory components and is frequently carried on conjugative plasmids, enhancing their potential for horizontal gene transfer [27].

The *vanA* operon remains the most reported *van* gene cluster in the United States and many European countries [28]. However, *vanB* has emerged as the predominant cluster in several regions, including Germany, Latvia, Denmark, the Netherlands, Poland, and Australia. In contrast, data from South America, Asia, and Africa remain limited, and further surveillance is needed to determine the dominant resistance operons in these regions [28].

The *vanA* and *vanB* clusters comprise a two-component regulatory system, in which the sensor kinase VanS and response regulator VanR detect the presence of vancomycin and initiate transcription of resistance genes. This results in the enzymatic replacement of D-Ala with D-lactate in the peptidoglycan precursor, significantly reducing vancomycin’s binding affinity (Figure 1) [25]. Consequently, cell wall synthesis proceeds unimpeded despite the presence of the antibiotic, enabling bacterial survival and continued propagation of resistance.

### 2.2. Behavioural Factors

Human behaviour plays a pivotal role in the emergence and spread of VREfm within healthcare environments. The overuse and inappropriate prescribing of broad-spectrum antibiotics, including vancomycin, cephalosporins, carbapenems, and fluoroquinolones, can disrupt the normal gut microbiota, creating an environment conducive to VREfm colonization and overgrowth [29,30,31,32]. Such dysbiosis not only facilitates colonization but also increases the risk of subsequent bloodstream infections, especially in vulnerable populations. In concert with broad-spectrum antibiotic use, the increased exposure to healthcare settings independently increases the risk of VREfm infection [31]. Although some studies implicate vancomycin directly as a risk factor for hospital-acquired VREfm [33,34,35,36], others find no significant association, likely due to methodological differences such as confounding by length of hospital stay and control group selection bias [37,38].

Beyond broad-spectrum antibiotic use, poor adherence to infection control protocols —such as inconsistent hand hygiene or inadequate environmental cleaning—facilitates cross-transmission. Invasive medical devices further compound the risk by providing surfaces for biofilm formation and bacterial persistence. Notably, a large proportion of *E. faecium* associated with device-related infections (e.g., central line-associated bloodstream infections [CLABSI] and catheter-associated urinary tract infections [CAUTI]) are vancomycin-resistant [32,39].

Due to widespread resistance to conventional agents with Gram-positive activity, such as aminoglycosides and β-lactams, vancomycin-resistant *E. faecium* (VREfm) is typically treated with last-line antibiotics including linezolid and daptomycin. However, resistance to both agents is increasingly reported, posing a serious threat to clinical management. Resistance to last-resort antibiotics such as daptomycin and linezolid may arise through direct, genetically encoded mechanisms that contribute significantly to treatment failure in *E. faecium*. Daptomycin is a cyclic lipopeptide that disrupts membrane integrity by binding to the cell membrane in a calcium-dependent manner, leading to depolarisation, ion leakage, and bacterial death in the absence of cell lysis [40]. However, mutations affecting the cell envelope stress response pathways (e.g., *liaFSR*, *yycFG*) and membrane phospholipid metabolism (e.g., *cls*, *gdpD*) can confer resistance by altering membrane composition and charge, thereby reducing daptomycin binding and activity [40,41]. Among these, *liaFSR* mutations are most observed and may be sufficient to reduce susceptibility, although additional mutations can amplify resistance [40,41]. These changes typically evolve under daptomycin selection pressure and may revert once antibiotic pressure is withdrawn [42,43,44].

Linezolid, a purely synthetic antibiotic of class oxazolidinone, inhibits bacterial protein synthesis by binding to the 23S rRNA of the 50S ribosomal subunit [45]. Linezolid is also subject to multiple resistance mechanisms. Acquired resistance most commonly involves point mutations in domain V of the 23S rRNA gene, particularly G2576T, which diminish linezolid binding affinity [46]. In addition, horizontally acquired genes such as those in the *cfr* family (encoding methyltransferases), as well as *optrA*, and *poxtA* (encoding ATP-binding cassette proteins) interfere with linezolid’s binding to the ribosomal subunit [40,47,48,49]. These genes are often plasmid-borne and capable of horizontal transfer, raising concern for broader dissemination [50,51]. Epidemiological studies show that linezolid resistance can arise both through selection in individual patients and through clonal spread during VRE outbreaks, particularly in high-consumption hospital settings [52]. Resistance levels also correlate with the number of mutated 23S gene copies, which may increase under prolonged therapy [53].

Aside from well-characterised mutational mechanisms, *E. faecium* can also develop resistance through unexpected, indirect pathways—including cross-resistance between unrelated antibiotic classes—further complicating antimicrobial stewardship efforts [54]. Rifaximin, a non-absorbable oral antibiotic commonly used as prophylaxis in hepatic encephalopathy and other gastrointestinal conditions, has been implicated in driving resistance to daptomycin—a structurally and mechanistically distinct last-resort antibiotic used to treat serious VREfm infections [54,55]. Despite its localised activity in the gut and its historical classification as low risk for resistance development, prolonged rifaximin exposure has been shown to select for mutations in the *rpoB* gene, which encodes the RNA polymerase β-subunit. Although *rpoB* is not directly linked to the bacterial membrane (the target of daptomycin), mutations activate the *prdRAB* operon, altering membrane charge and reducing daptomycin binding [54]. This cross-resistance mechanism has important clinical implications. In patients previously exposed to rifaximin, daptomycin may be less effective, increasing the risk of treatment failure in VREfm infection. ASPs should therefore consider reviewing rifaximin prescribing policies, especially in high-risk patient populations such as ICU or those with recurrent hepatic encephalopathy. This mechanism of daptomycin resistance has been documented in global VREfm isolates [54], challenging the assumptions that underpin conventional antimicrobial stewardship and illustrating the complexity of microbial adaptation under selective pressure.

### 2.3. Social, Economic, and Environmental Factors

The emergence and persistence of VREfm are shaped by interconnected social, economic, and environmental drivers. Historically, the use of the glycopeptide avoparcin in livestock feed across Europe exerted selective pressure for VanA-type VRE, which were subsequently isolated from meat products and healthy individuals in the community. After avoparcin was banned, a marked decline in VRE carriage was observed in both poultry and the human gut microbiome, underscoring the role of agricultural antibiotic use in community-level resistance [56].

Environmental contamination is another key factor. VREfm has been frequently detected in wastewater, rivers, and even treated effluent, indicating that antibiotic residues and resistant bacteria escape conventional treatment systems [57]. Alarmingly, hospital-adapted VREfm clones have been identified in municipal sewage, highlighting environmental reservoirs as vectors for reintroducing resistant strains into human populations [58]. These findings underscore the necessity for integrated surveillance approaches and improved waste management practices under the One Health framework, which recognises the interconnectedness of human, animal, and environmental health [59].

## 3. Social and Global Disparities in VREfm Infections

The burden of VREfm infections is shaped by global disparities in healthcare infrastructure, antibiotic stewardship practices, and surveillance capacity, as well as by local social and economic inequalities affecting vulnerable patient groups. Together, these factors contribute to the uneven distribution of VREfm prevalence and outcomes across the world.

In high-income countries (HICs), particularly across Europe, North America, and Australia, widespread use of vancomycin and broad-spectrum antibiotics has driven an increase in VREfm prevalence [60]. Surveillance data from the UK Health Security Agency show that approximately 21% of *E. faecium* bloodstream isolates are now vancomycin-resistant across the UK [4]. Similarly, in the United States, vancomycin-resistant *Enterococcus* is classified as a major antimicrobial resistance (AMR) threat, responsible for 54,500 hospital-acquired infections and 5400 deaths in 2017 [5]. The rate of VREfm bacteraemia has risen steadily in the UK by 63.5% between 2013 and 2021 [4], and a comparable rise has been observed in the United States, with 20,000 HAI cases and 1300 deaths attributed to VREfm in 2013 [61].

By contrast, reported rates of VREfm in low- and middle-income countries (LMICs) tend to be lower, although this likely reflects underreporting rather than a truly reduced burden of disease. Many LMICs lack comprehensive AMR surveillance networks, particularly in rural and secondary hospitals [62]. Moreover, widespread unregulated access to antibiotics, as seen in countries like India, has contributed significantly to AMR. Antibiotics are often available over the counter without prescription, and empirical broad-spectrum antibiotic use remains common, even for non-bacterial infections [62]. Although national AMR strategies such as India’s National Action Plan are under development, implementation remains inconsistent and fragmented [63]. Without strong antimicrobial stewardship programmes to guide rational antibiotic use, selective pressure for VREfm and other resistant organisms continues to grow.

Within healthcare systems worldwide, certain patient groups face disproportionate risks of VREfm colonisation and invasive infection. Vulnerable populations such as cancer patients, transplant recipients, and critically ill ICU patients are particularly at risk. Invasive devices such as central venous catheters, ventilators, and urinary catheters provide surfaces for biofilm formation and bacterial persistence. In the United States, *E. faecium* is among the top pathogens associated with CLABSI, with three out of four CLABSI cases caused by vancomycin-resistant strains [64]. Similarly, UK data show that males aged over 75 years have the highest rate of *Enterococcus* spp. bacteraemia among demographic groups [4].

Healthcare system inequities further exacerbate these risks. Under-resourced hospitals, particularly in LMICs, often experience staffing shortages, inadequate infection control measures, and limited access to effective alternative therapies such as linezolid and daptomycin [63]. Socioeconomically disadvantaged patients are more likely to experience delays in diagnosis, suboptimal treatment options, and increased exposure to environments conducive to the spread of resistant organisms. Addressing these disparities is essential for designing effective, equitable strategies to curb the global burden of VREfm.

## 4. Mapping Prevention Activities Across the Disease Pathway

Effective control of VREfm requires a multifaceted prevention strategy, targeting different stages of the disease pathway. Prevention activities can be conceptualised within a framework of primary, secondary, and tertiary prevention, each addressing a distinct point from pathogen emergence to clinical impact (Figure 2).

### 4.1. Primary Prevention (Preventing Emergence and Spread)

Primary prevention focuses on reducing the opportunities for VREfm to emerge and disseminate within healthcare settings. ASPs are a cornerstone intervention, aiming to restrict the inappropriate use of vancomycin and other broad-spectrum antibiotics that drive resistance [28,65]. By developing and enforcing prescribing guidelines—particularly for high-risk groups such as haematology and ICU patients—ASP can reduce unnecessary antibiotic exposure and selective pressure [54,66].

Alongside pathogen-specific stewardship measures, robust “universal” hospital infection control interventions are critical. Enhancing hand hygiene compliance among healthcare workers has been shown to significantly reduce VREfm transmission [67,68]. The value of isolating colonised patients remains controversial, with limited evidence supporting isolation as an effective measure to reduce VRE infections [69,70]. Environmental cleaning protocols also play a vital role; frequent decontamination of hospital surfaces and medical equipment with disinfectants effective against VRE is necessary, given the organism’s ability to survive for extended periods on dry surfaces [71]. This highlights the need for horizontal infection prevention measures to complement pathogen-specific ASP interventions [28].

### 4.2. Secondary Prevention (Early Detection and Containment)

Secondary prevention aims to identify VREfm colonisation or infection early to contain its spread. Active surveillance screening, particularly in high-risk groups such as ICU admissions, transplant recipients, and oncology patients, has been promoted as a cost-effective strategy to enable early detection, appropriate antimicrobial therapy, and containment [72,73]. However, other studies have questioned the clinical value of active surveillance.

A large cluster-randomised trial involving 74 ICUs and more than 70,000 patients reported enhanced effectiveness of universal decolonisation without screening compared to targeted decolonisation guided by screening [71]. Similarly, a retrospective cohort study from a Danish hospital found no significant difference in the number of bacteraemia cases, 30-day mortality, or deaths attributable to VREfm after discontinuing screening and isolation protocols [74]. However, this study’s relatively short follow-up period limits definitive conclusions. Complementary evidence shows that although VRE screening identifies colonised patients, it provides limited guidance for treatment decisions. In a 280-bed tertiary-care hospital in the United States, a retrospective study found that while a positive VRE screen increased the risk of infection thirteenfold, the absolute risk of developing infection was only 13%, whereas a negative screen had a 98% negative predictive value [75]. Taken together, this evidence suggests that routine active surveillance screening for VREfm colonisation may not provide substantial clinical benefit over universal decolonisation strategies. As such, routine screening is not currently recommended in all settings. However, further large-scale, context-specific studies are needed to refine best practices for specific patient populations, such as ICU, transplant, and oncology patients.

Emerging strategies such as the use of prophylactic probiotics to reduce gut colonisation with VREfm are also under investigation [29]. In animal models, administration of *Lactobacillus* spp. significantly decreased VRE colonisation [76,77,78], but trials in human patients have been limited, often underpowered, and have shown inconsistent efficacy [79,80,81]. Furthermore, concerns exist that certain probiotic strains could facilitate horizontal gene transfer of vancomycin resistance within the gut microbiome [82].

### 4.3. Tertiary Prevention (Minimising Complications and Impact)

Tertiary prevention seeks to minimise the clinical consequences of VREfm infections once they have occurred. Treatment options remain limited due to the intrinsic resistance of *E. faecium* to many antibiotics and the increasing prevalence of acquired resistance mechanisms. Optimising treatment regimens, including the use of newer antimicrobial agents such as oritavancin, dalbavancin, and tigecycline in cases where conventional therapies fail, can improve patient outcomes [83,84,85,86].

Beyond single-agent therapies, there is increasing interest in the use of combination regimens, especially involving daptomycin and β-lactams, as a strategy to enhance efficacy and overcome resistance in *E. faecium*. Daptomycin is a last line lipopeptide antibiotic that targets the bacterial membrane, but resistance can emerge rapidly, often due to alterations in membrane structure and surface charge [42]. However, these adaptations may incur fitness costs and induce, whereby certain daptomycin-resistant strains become resensitised to glycopeptides like vancomycin due to disruptions in the *vanA* operon [42].

Combining daptomycin with β-lactams, including ampicillin, ceftaroline, ceftriaxone, ertapenem, or fosfomycin, has been shown in both in vitro and clinical settings to synergistically enhance bactericidal activity, even against daptomycin-non-susceptible (DNS) or vancomycin-resistant isolates [87,88,89]. These effects are primarily attributed to β-lactam-induced reductions in cell surface charge and increased membrane permeability, which facilitate daptomycin binding and improve killing [88,89]. Other combination strategies are under investigation. Tigecycline with high-dose daptomycin or gentamicin may be especially effective in endocarditis or refractory bacteraemia [90,91,92]. Similarly, resistance to daptomycin monotherapy may be overcome by the co-administration of linezolid and doxycycline [93,94].

Beyond conventional antibiotics, novel strategies such as bacteriophage therapy and phage-derived enzymes are under investigation. These therapies offer highly specific mechanisms to eradicate VREfm with or without adjunct antibiotic therapy [95]. In a longitudinal case study, adjunctive phage therapy targeting *E. faecium* resulted in reduced intestinal burden, improved symptom control, and delayed recurrence of bloodstream infection when added to failing antibiotic regimens [96]. In vitro testing confirmed that the combination of phage and antibiotic was more suppressive than either alone. However, clinical efficacy may be limited by the emergence of anti-phage neutralising antibodies, underscoring the need for continued monitoring and refinement of this approach [96]. Experimental vaccines and monoclonal antibodies targeting *E. faecium* virulence factors are also being explored, although significant challenges remain, particularly in immunocompromised populations [97].

Despite the promise of these tertiary strategies, reliance on reactive treatments alone is insufficient. The slow pace of new drug development, the risk of resistance to last-line agents, and the complexity of hospital-acquired infections all reinforce the need for strong upstream preventive strategies, including enhanced antimicrobial stewardship, robust surveillance, and reinforced infection control measures.

## 5. The Evolving Role of Antimicrobial Stewardship Programmes (ASP) in Preventing VREfm

ASPs remain a cornerstone of strategies to prevent VREfm infections. By targeting the inappropriate use of vancomycin and other broad-spectrum antibiotics, ASPs reduce the selective pressure that drives resistance and help preserve the effectiveness of critical antimicrobials such as linezolid [66,98].

Broad-spectrum antibiotic use is a major behavioural driver of VREfm colonisation and subsequent infection. Agents such as carbapenems, cephalosporins, fluoroquinolones, and vancomycin can profoundly disrupt the gut microbiota, creating ecological niches that facilitate VREfm overgrowth. This is particularly relevant in elderly or immunocompromised patients, who may be colonised asymptomatically but remain at risk for bloodstream infection during hospitalisation or invasive procedures [30,31,32]. Antimicrobial stewardship programmes therefore play a critical role in reducing unnecessary exposure to these high-risk antibiotics, thereby limiting both colonisation pressure and progression to invasive disease. ASPs also contribute to risk stratification and the rational de-escalation of empirical therapy in patients known to be colonised with VREfm, further aligning treatment with microbiological risk and preserving narrow-spectrum options.

A leading example of a national stewardship framework is Australia’s Antimicrobial Use and Resistance in Australia (AURA) Surveillance System, coordinated by the Australian Commission on Safety and Quality in Health Care [99]. AURA integrates data from laboratory surveillance (including the Australian Enterococcal Surveillance Outcome Program [AESOP]), alert systems (CARAlert: National Alert System for Critical Antimicrobial Resistances), and passive surveillance (APAS: Australian Passive AMR Surveillance). In 2023, AESOP recorded 1599 episodes of enterococcal bacteraemia, with *E. faecium* responsible for 41.1% of cases, an increase of 10.2% from 2022. The rate of vancomycin-resistant *E. faecium* (VREfm) rose from 46.9% in 2022 to 50.8% in 2023, with 53.2% of isolates carrying *vanA* and/or *vanB* genes. Clinical implications were severe: the 30-day all-cause mortality rate for *E. faecium* was 26.3%, and over 23% of patients experienced a hospital stay exceeding 30 days [99]. These findings underscore the growing burden of VREfm and the need for responsive, data-informed antimicrobial stewardship.

In England, the ‘Start Smart Then Focus’ (SSTF) initiative represents another effective national ASP model. SSTF provides structured guidance for timely antibiotic review (48–72 h after initiation) with clear documentation of clinical indication and treatment duration [100,101]. Antimicrobial prescribing is then tailored to the patient based on clinical response, microbiology results, and local resistance data. Surveys show SSTF implementation across >90% of acute trusts [102], though variability remains in audit uptake and multidisciplinary engagement. Dedicated antimicrobial stewardship committees and specialist pharmacists play a pivotal role, particularly in secondary care. SSTF also promotes community pharmacy involvement and emphasises professional education as part of the UK’s AMR strategy [101]. In primary care—where the majority of antibiotics are prescribed in the UK—the TARGET (Treat Antibiotics Responsibly, Guidance, Education, Tools) Antibiotics toolkit provides a comprehensive, evidence-based approach to engaging general practitioners (GPs) in antimicrobial stewardship. Developed by the Royal College of General Practitioners in collaboration with the Antimicrobial Stewardship in Primary Care (ASPIC) group, TARGET equips clinicians with practical resources to support responsible prescribing [103]. These include educational materials for clinicians and patients, audit tools, and consultation aids that encourage GPs to use routine appointments as opportunities to raise awareness about antimicrobial resistance (AMR). One key strategy is the use of back-up (delayed) antibiotic prescriptions, which have been shown to reduce unnecessary antibiotic consumption without compromising symptom resolution. By providing a prescription to be used only if symptoms worsen or persist, GPs can support patient self-management, reduce re-consultations, and help prevent complications as effectively as immediate antibiotic treatment.

Beyond SSTF and TARGET, UK national policy has reinforced AMS through the 2013 Antimicrobial Prescribing and Stewardship (APS) Competency Framework, which defines and supports clinician training, specifically on the responsible prescription of antimicrobials [104]. Complementing this, UK efforts to de-label inappropriate penicillin allergies—found in ~10% of the population but confirmed in <1%—are being scaled up to preserve β-lactams and reduce unnecessary use of broader-spectrum agents such as carbapenems and fluoroquinolones [105]. At the strategic level, the UK’s 5-Year National Action Plan for AMR (2024–2029) adopts a One Health approach to reduce unnecessary antimicrobial use across human, animal, and environmental sectors. Since 2014, human exposure to antimicrobials has fallen by over 8%, despite pressures from the COVID-19 pandemic and Group A streptococcal outbreaks. In parallel, sales of highest-priority critically important antimicrobials in food-producing animals have declined by 81% between 2014 and 2022 [106]. These comprehensive national approaches provide a valuable contrast to the challenges faced by many LMICs, where fragmented policies, limited diagnostic capacity, and insufficient surveillance infrastructure hinder the development of robust stewardship systems. Compared with containment measures alone, stewardship offers a proactive approach that encompasses primary, secondary, and tertiary prevention, contributing to longer-term control of AMR.

In addition to improving clinical outcomes, ASPs have been associated with cost savings through reduced infection rates, shortened hospital stays, and more efficient antibiotic usage. Their multidisciplinary nature—typically involving infectious disease specialists, pharmacists, microbiologists, and infection control professionals—enhances their effectiveness [107].

However, emerging research has challenged some traditional assumptions underpinning stewardship strategies. Specifically, the unexpected emergence of cross-resistance between rifaximin and daptomycin in VREfm highlights that antimicrobial exposure, even to agents seemingly unrelated to a target pathogen, can select for critical resistance traits [54]. This phenomenon, driven by surprising genetic alterations affecting the bacterial cell membrane, underscores the complexity of resistance evolution and the limitations of focusing solely on direct antibiotic-pathogen relationships.

Future stewardship models must broaden their scope. Rather than simply restricting the use of antibiotics directly linked to VREfm emergence, stewardship efforts should incorporate a deeper understanding of collateral resistance mechanisms, systematic surveillance of emerging resistance patterns, and critical evaluation of all antimicrobial use—including prophylactic regimens previously considered low-risk. Only through such adaptive, forward-thinking strategies can ASPs continue to play a central role in containing VREfm and preserving the efficacy of last-resort antimicrobials.

## 6. Limitations and Future Directions

Despite substantial progress in understanding and responding to VREfm, key challenges remain across biological, clinical, and stewardship domains. This section outlines these limitations and proposes future directions to inform a more comprehensive response.

### 6.1. Biological Challenges

The genetic plasticity of *E. faecium*—including its ability to acquire and disseminate resistance genes such as *vanA*, *vanB*, *optrA*, and *poxtA*—poses an ongoing threat to treatment efficacy [26,47,48]. The rapid emergence of resistance during therapy, particularly to daptomycin and linezolid [22,23,24], limits the clinical utility of these last-resort agents. Biofilm formation remains clinically relevant in persistent infections and device-related colonisation and is associated with increased antimicrobial tolerance [12,20]. Future work should address the role of biofilms in facilitating resistance acquisition, particularly in the context of indwelling medical devices and invasive procedures. Further research is required to develop a more complete mechanistic understanding of the bacterial genes facilitating survival on inanimate surfaces, and thus persistence and transmission in the hospital environment.

### 6.2. Clinical Management Challenges

Treatment options for VREfm remain constrained, particularly due to the ability of clinical strains to adapt and acquire mutations and/or mobile genetic elements encoding antibiotic resistance genes. While combination therapies (e.g., daptomycin plus β-lactams or linezolid and doxycycline) show promise, robust clinical trial data validating their efficacy are lacking, with much of the evidence being accrued from isolated case reports [87,89]. Although phage therapy and microbiome modulation strategies offer novel avenues for treatment or decolonisation, most supporting evidence is limited to case reports, in vitro models, or small trials [80,96]. Other members of the gut microbiota can contribute significantly to suppressing or displacing VREfm colonisation. Harnessing these natural competitive interactions could inform the development of affordable, scalable strategies to prevent or reduce VREfm carriage [97,108]. Larger, controlled studies are needed to evaluate the clinical impact, safety, and long-term efficacy of these emerging therapeutic modalities.

### 6.3. Stewardship Gaps, Surveillance Disparities, and Implementation Challenges

In addition to biological and clinical uncertainties, key challenges persist around antimicrobial stewardship implementation, particularly in resource-limited settings. Despite the development of coordinated national programmes in countries such as the UK and Australia [99,100], the global success of stewardship hinges on adaptable models that account for local infrastructure, diagnostic capacity, and prescribing norms. In many LMICs, limited access to microbiological diagnostics and specialist personnel constrains real-time prescribing oversight, while antibiotics are often dispensed without adequate regulation [63,109]. Tailored stewardship strategies that are low-cost, scalable, and compatible with minimal laboratory support are urgently needed to bridge this implementation gap [109]. Equally, overemphasis on restricting specific antimicrobial classes may overlook broader patterns of collateral resistance, such as the emergence of daptomycin resistance following rifaximin prophylaxis, highlighting the need for stewardship programmes to move beyond simple pathogen–antibiotic pairings and account for complex resistance ecologies [54,55]. In this context, expanding research into cost-effective interventions, namely decision support tools, context-specific prescribing guidelines, and community-level education, should be a priority.

At the same time, global AMR surveillance systems remain patchy and fragmented. Much of the available epidemiological data comes from high-income countries or large tertiary centres, limiting our understanding of VREfm prevalence, resistance patterns, and transmission in rural, secondary, or community-based settings [62,110]. Strengthening integrated surveillance capacity in LMICs is essential to detect emerging resistance trends early, inform stewardship efforts, and improve equity in global AMR control [110].

In addition, future policy and research efforts should be grounded in a One Health framework, which recognises the interdependence of human, animal, and environmental health [59]. Integrated surveillance systems and stewardship approaches that account for antibiotic use in agriculture, water contamination, and hospital transmission dynamics will be essential for long-term containment of VREfm [110].

Finally, while many stewardship interventions have shown promise in controlled environments, their long-term effectiveness and sustainability across diverse healthcare systems remain poorly understood. Further research is needed to evaluate real-world outcomes of stewardship programmes, ideally through pragmatic trials and health systems research that reflect the heterogeneity of clinical practice across settings.

## 7. Conclusions

The spread of VREfm is driven by a complex interplay of biological, behavioural, and healthcare system factors. Global and local health inequalities, particularly among immunocompromised patients and in resource-limited settings, further amplify the burden of VREfm infections. ASPs remain a critical component of prevention strategies, addressing the root causes of resistance by optimising antibiotic use and preserving the efficacy of last-resort agents.

However, emerging evidence of unexpected cross-resistance mechanisms challenges conventional stewardship models and highlights the need for more adaptive, evidence-driven strategies. Strengthening stewardship efforts must go hand-in-hand with broader infection control interventions, environmental management, and innovative treatment development.

Looking forward, continued research into novel prevention and treatment approaches—including microbiome-based therapies, phage applications, and new antimicrobials—is essential. Strengthening global surveillance networks, embedding stewardship principles into healthcare policy, and addressing disparities in healthcare access and quality will be crucial for limiting the spread of VREfm. Ultimately, a coordinated, multidisciplinary, and equitable approach is needed to effectively confront this growing antimicrobial resistance threat.

## Figures and Tables

**Figure 1 antibiotics-14-00522-f001:**
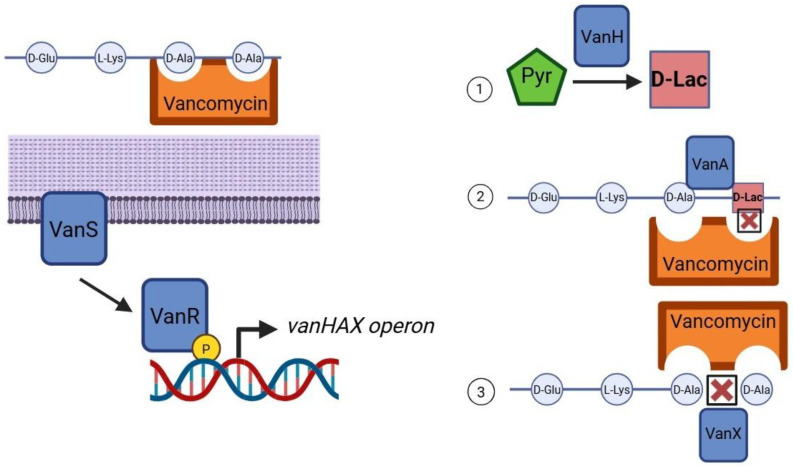
Mechanism of vancomycin resistance in *Enterococcus faecium*. This schematic illustrates the induction and effect of vancomycin resistance mediated by the *vanA* or *vanB* operon. On the left, vancomycin binds to the D-Ala–D-Ala terminus of the peptidoglycan precursor, preventing cross-linking and inhibiting cell wall synthesis. Detection of vancomycin by the membrane-bound sensor kinase VanS leads to autophosphorylation and phosphate transfer to the cytoplasmic response regulator VanR, which activates transcription of the *vanHAX* operon. On the right, the enzymatic products of this operon are shown: (1) VanH reduces pyruvate to D-lactate (D-Lac); (2) VanA ligates D-Ala to D-Lac, producing the D-Ala–D-Lac dipeptide; and (3) VanX cleaves residual D-Ala–D-Ala, ensuring exclusive incorporation of the resistant precursor. Vancomycin is unable to bind to D-Ala–D-Lac, thus permitting normal peptidoglycan cross-linking and conferring high-level resistance. Created with BioRender.com.

**Figure 2 antibiotics-14-00522-f002:**
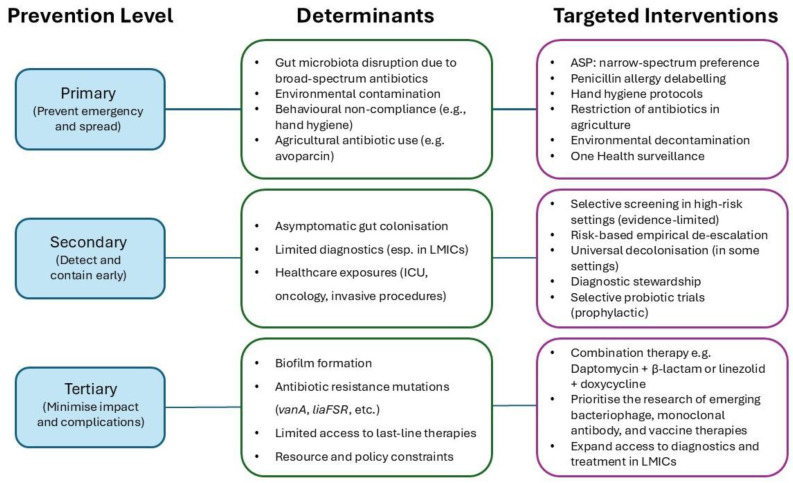
Conceptual framework linking determinants, levels of prevention, and targeted interventions for VREfm control. This schematic integrates the key biological, behavioural, and systemic determinants driving the emergence and spread of vancomycin-resistant *Enterococcus faecium* (VREfm). These determinants are mapped onto three levels of prevention: primary (limiting emergence and transmission), secondary (facilitating early detection and containment), and tertiary (reducing clinical impact and complications). Corresponding evidence-based interventions are aligned to each level, including: antimicrobial stewardship programmes, strategic de-escalation of high-risk antibiotic use in clinical and agricultural settings, enhanced surveillance in low- and middle-income countries (LMICs), and the development of novel and combination treatment strategies. This framework highlights the importance of integrated, multidisciplinary responses to effectively contain VREfm at local, national, and global levels.

## Data Availability

No new data were created.

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
