# Peer review of "Vancomycin-Resistant E. faecium: Addressing Global and Clinical Challenges"

_antibiotics, 2025, doi:10.3390/antibiotics14050522_

Round 1
Reviewer 1 Report
Comments and Suggestions for Authors
The manuscript entitled "Vancomycin-Resistant E. faecium: Addressing Global and Clinical Challenges" provides a clear, up-to-date, and relevant overview of the biological, clinical, and public health dimensions of vancomycin-resistant Enterococcus faecium (VREfm) infections.
The article is structured as a mini-review, which, in my opinion, is inconsistent with the format of a review.
Please consider the following recommendations:
Lines 7–21: I recommend including one or two specific examples of "emerging threats" (e.g., cross-resistance) to enhance the clarity and appeal of the summary.
Lines 52–83:
- The explanation of the vanA and vanB clusters is solid. I suggest adding a simplified graphical representation (e.g., diagram with the D-Ala → D-Lac substitution mechanism).
- Optional: add a brief mention of the global distribution of vanA vs. vanB genotypes, if relevant data are available.
Lines 84–111: The inclusion of the rifaximin-induced cross-resistance mechanism is excellent. Additional commentary on its clinical relevance (e.g., risk of treatment failure, changes in usage recommendations) would be useful.
Lines 112–127: You mentioned the "One Health" framework, but do not return to it later. I suggest revisiting the concept in the conclusions or future directions.
Lines 193–217: Divergent opinions on active screening are presented. A clearer conclusion would be helpful: do the authors recommend routine screening in ICUs or other wards?
Lines 234–260: I suggest including a concrete example of an effective national ASP program (e.g., AURA in Australia or START Back in the UK) to provide a counterpoint to the challenges in LMICs.
Figures/Tables: Completely missing. The manuscript would benefit from a summary diagram or conceptual framework integrating: Determining factors (biological, behavioral, systemic), Levels of prevention (primary, secondary, tertiary), Proposed interventions (e.g., stewardship, surveillance, new therapies).
Author Response
The manuscript entitled "Vancomycin-Resistant E. faecium: Addressing Global and Clinical Challenges" provides a clear, up-to-date, and relevant overview of the biological, clinical, and public health dimensions of vancomycin-resistant Enterococcus faecium (VREfm) infections.
The article is structured as a mini-review, which, in my opinion, is inconsistent with the format of a review.
Please consider the following recommendations:
Lines 7–21: I recommend including one or two specific examples of "emerging threats" (e.g., cross-resistance) to enhance the clarity and appeal of the summary.
Thank you for this suggestion. We have revised the abstract to explicitly mention the recently discovered cross-resistance between rifaximin and daptomycin as a key emerging threat. This addition enhances the clarity and highlights the clinical implications of such unexpected mechanisms of resistance.
Lines 52–83:
- The explanation of the vanA and vanB clusters is solid. I suggest adding a simplified graphical representation (e.g., diagram with the D-Ala → D-Lac substitution mechanism).
- Optional: add a brief mention of the global distribution of vanA vs. vanB genotypes, if relevant data are available.
We appreciate these helpful recommendations. We have now included a simplified diagram illustrating the key biochemical mechanism underlying vancomycin resistance (the substitution of D-Ala-D-Ala with D-Ala-D-Lac), which visually clarifies this critical concept (Figure 1).
Additionally, we have briefly commented on the global predominance of vanA vs. vanB genotypes, citing recent literature that demonstrates vanA clusters as more prevalent globally, but data on LMICs is lacking.
Lines 84–111: The inclusion of the rifaximin-induced cross-resistance mechanism is excellent. Additional commentary on its clinical relevance (e.g., risk of treatment failure, changes in usage recommendations) would be useful.
Thank you for highlighting this important point. We have expanded this section to specifically discuss the clinical implications, including the increased risk of daptomycin treatment failure due to rifaximin-induced resistance. We have also added commentary suggesting that stewardship programmes may need to reconsider rifaximin use guidelines, particularly in high-risk patient groups.
Lines 112–127: You mentioned the "One Health" framework, but do not return to it later. I suggest revisiting the concept in the conclusions or future directions.
Thank you for pointing out this oversight. We have now explicitly revisited the "One Health" framework in section 6, limitations and future directions, emphasising the need for integrated surveillance and stewardship efforts spanning human, animal, and environmental sectors to effectively manage VREfm.
Lines 193–217: Divergent opinions on active screening are presented. A clearer conclusion would be helpful: do the authors recommend routine screening in ICUs or other wards?
We agree that a clear recommendation would improve clarity. We have added a specific recommendation indicating that based on current evidence, routine universal screening for VREfm colonisation in ICUs may not offer significant clinical benefit and that targeted interventions or universal decolonisation strategies may be preferable. We also emphasise the need for further large-scale, context-specific studies to refine these recommendations.
Lines 234–260: I suggest including a concrete example of an effective national ASP program (e.g., AURA in Australia or START Back in the UK) to provide a counterpoint to the challenges in LMICs.
Thank you for this valuable suggestion. We have incorporated a brief description of Australia's AURA (Antimicrobial Use and Resistance in Australia) programme as a successful example of a national ASP framework. We use this to highlight effective stewardship strategies, providing a useful contrast to the described challenges faced by resource-limited settings.
Figures/Tables: Completely missing. The manuscript would benefit from a summary diagram or conceptual framework integrating: Determining factors (biological, behavioral, systemic), Levels of prevention (primary, secondary, tertiary), Proposed interventions (e.g., stewardship, surveillance, new therapies).
We have now included a comprehensive conceptual summary diagram integrating the biological, behavioural, and systemic determinants of VREfm, alongside the different levels of prevention and proposed interventions (Figure 2). This figure significantly enhances the manuscript's clarity and facilitates a better understanding of the complex interplay between these elements.
Reviewer 2 Report
Comments and Suggestions for Authors
- Please improve introduction section with the recent publications and advantages of this review paper.
- Author describe limitations in this direction but I suggest to extend the limitation section with structured subheadings that focus on specific challenges with VREFfm.
Author Response
- Please improve introduction section with the recent publications and advantages of this review paper.
Thank you for this suggestion. We have enhanced the introduction by including additional recent publications to better contextualise the importance and timeliness of this review. The first reference is purposefully historical to explicitly indicate how AMR is not a new phenomenon nor a new discovery. Moreover, we have now explicitly highlighted the advantages of our review paper, emphasising how it synthesises recent advances and clarifies key emerging issues in the field.
- Author describe limitations in this direction but I suggest to extend the limitation section with structured subheadings that focus on specific challenges with VREFfm.
We appreciate this constructive feedback. We have expanded the limitations section by introducing structured subheadings that systematically address specific challenges associated with VREfm. These subheadings now clearly delineate biological, clinical, and stewardship-related limitations, which substantially enhance the clarity and comprehensiveness of this section.
Reviewer 3 Report
Comments and Suggestions for Authors
This manuscript is a literature review by Radford-Smith and Anthony summarizing our current understanding of the spread of Vancomycin-resistant Enterococcus faecium and available preventative measures. This review is important because of the overall danger of E. faecium infections and the knowledge on prevention strategies will limit the spread of infections. This review is clear, well-written, and thoughtful. My comments are intended to improve the clarity and accessibility of this review.
Comments:
- Section 2.1. In enterococcus, E. faecalis are the major biofilm formers while it is quite rare for E. faecium, as evidenced by reference #12 in manuscript. Adding more primary literatures about E. faecium that are biofilm formers and the isolation of persister E. faecium cells will strengthen this section.
- Section 2.2. The authors discussed the resistance mechanism against daptomycin as a result of rifaximin prophylaxis. However, other daptomycin or linezolid resistance mechanisms should also be explored, as these antibiotics are part of the last-line therapeutics.
- Line 139 – 140. According to the CDC’s Antibiotic Resistance Threats in the United States, published in 2017, VREfm is responsible for 54,500 hospital acquired infections and 5,400 deaths annually. Please update this sentence accordingly.
- Section 5. Prior literatures have shown that resistance to daptomycin may re-sensitize faecium to ampicillin and its derivatives, showing the effect of combinatorial therapy. The merits of this strategy should be discussed
- Section 5. Rifaximin prophylaxis increases daptomycin resistance incidences in faecium. However, prior literatures suggest that the primary method by which VREfm causes bloodstream infections is due to the spread of VREfm from gut after broad-spectrum antibiotics decimated other competing gut commensals. The use and role of other broad-spectrums should also be discussed.
Author Response
This manuscript is a literature review by Radford-Smith and Anthony summarizing our current understanding of the spread of Vancomycin-resistant Enterococcus faecium and available preventative measures. This review is important because of the overall danger of E. faecium infections and the knowledge on prevention strategies will limit the spread of infections. This review is clear, well-written, and thoughtful. My comments are intended to improve the clarity and accessibility of this review.
Comments:
- Section 2.1. In enterococcus, E. faecalis are the major biofilm formers while it is quite rare for E. faecium, as evidenced by reference #12 in manuscript. Adding more primary literatures about E. faecium that are biofilm formers and the isolation of persister E. faecium cells will strengthen this section.
Thank you for highlighting this point. We have now clarified the relative rarity of biofilm formation in E. faecium compared to E. faecalis, and we have included additional primary literature specifically demonstrating biofilm formation and persister cell isolation in E. faecium. This provides a more accurate and robust representation of biofilm-related mechanisms in this pathogen.
- Section 2.2. The authors discussed the resistance mechanism against daptomycin as a result of rifaximin prophylaxis. However, other daptomycin or linezolid resistance mechanisms should also be explored, as these antibiotics are part of the last-line therapeutics.
We appreciate this important recommendation. We have expanded this section by incorporating other key resistance mechanisms to linezolid and daptomycin beyond the rifaximin-induced cross-resistance. Specifically, we have included a concise overview of genetic mutations and adaptive responses that contribute to linezolid and daptomycin resistance, reinforcing the section's comprehensiveness and clinical relevance.
- Line 139 – 140. According to the CDC’s Antibiotic Resistance Threats in the United States, published in 2017, VREfm is responsible for 54,500 hospital acquired infections and 5,400 deaths annually. Please update this sentence accordingly.
Thank you for pointing out the outdated reference. We have updated this sentence to accurately reflect the CDC’s Antibiotic Resistance Threats report (2017), noting that VREfm is responsible for approximately 54,500 hospital-acquired infections and 5,400 deaths annually.
- Section 5. Prior literatures have shown that resistance to daptomycin may re-sensitize faecium to ampicillin and its derivatives, showing the effect of combinatorial therapy. The merits of this strategy should be discussed
This is an insightful point. We have now incorporated a brief discussion highlighting the merit and clinical potential of combinatorial therapies, particularly the phenomenon by which daptomycin resistance can resensitise E. faecium to beta-lactams such as ampicillin. This has been added to section 4.3. Tertiary Prevention (Minimising Complications and Impact), as it discusses optimising treatment regimens once VREfm has already occurred.
- Section 5. Rifaximin prophylaxis increases daptomycin resistance incidences in faecium. However, prior literatures suggest that the primary method by which VREfm causes bloodstream infections is due to the spread of VREfm from gut after broad-spectrum antibiotics decimated other competing gut commensals. The use and role of other broad-spectrums should also be discussed.
We agree this is an important addition. In section 2.2 we originally wrote that “The overuse and inappropriate prescribing of broad-spectrum antibiotics, including vancomycin, can disrupt normal gut microbiota, creating conditions that promote colonisation by VREfm [20].” We have expanded the discussion to explicitly address the role of broad-spectrum antibiotics, beyond rifaximin, in promoting gut dysbiosis and facilitating the spread of VREfm.
In section 5 we highlight the role antibiotic stewardship plays in mitigating VREfm dissemination effectively.